Antioxidant status of rats’ blood and liver affected by sodium selenite and selenium nanoparticles

Urbankova Lenka 1
Horky Pavel 1
Skladanka Jiri 1
Pribilova Magdalena 1
Smolikova Vendula 2 4
http://orcid.org/0000-0002-8616-1344 Nevrkla Pavel 3
Cernei Natalia 2 4
Lackova Zuzana 2 4
Hedbavny Josef 2
Ridoskova Andrea 2 4
Adam Vojtech 2 4
http://orcid.org/0000-0003-4216-9544 Kopel Pavel 2 4 pavel.kopel@mendelu.cz
1 Department of Animal Nutrition and Forage Production, Faculty of AgriSciences, Mendel University in Brno , Brno , Czech Republic
2 Department of Chemistry and Biochemistry, Faculty of AgriSciences, Mendel University in Brno , Brno , Czech Republic
3 Department of Animal Breeding, Faculty of AgriSciences, Mendel University in Brno , Brno , Czech Republic
4 Central European Institute of Technology, Technical University of Brno , Brno , Czech Republic
Uversky Vladimir
Electronic publication date: 2018 May 28
Publication date: 2018
Volume: 6
Electronic Location ID: e4862
Received 2018 Feb 19; Accepted 2018 May 7
Copyright: © 2018 Urbankova et al.
Copyright year: 2018
Copyright holder: Urbankova et al.
License: This is an open access article distributed under the terms of the Creative Commons Attribution License, which permits unrestricted use, distribution, reproduction and adaptation in any medium and for any purpose provided that it is properly attributed. For attribution, the original author(s), title, publication source (PeerJ) and either DOI or URL of the article must be cited.
License URL: https://creativecommons.org/licenses/by/4.0/

Keywords: Selenium nanoparticles, Glutathione, Rat, Animal nutrition, Antioxidant

Funding: IP IGA MENDELU 17/2017 CEITEC 2020 (LQ1601) Ministry of Education, Youth and Sports of the Czech Republic under the National Sustainability Programme II The study was supported by IP IGA MENDELU 17/2017. Moreover, this research was carried out under the project CEITEC 2020 (LQ1601) with financial support from the Ministry of Education, Youth and Sports of the Czech Republic under the National Sustainability Programme II. The funders had no role in study design, data collection and analysis, decision to publish, or preparation of the manuscript.

==============================
Background

Selenium is an essential element; however, at higher doses, it can be toxic. Therefore, alternative nanotechnological solutions are required to overcome toxicological issues, rather than conventional alternatives. Nanoparticles show new and promising properties that may be able to suppress toxicity while maintaining the positive effects of selenium on an organism. The aim of the experiment was to determine the influence of sodium selenite and selenium nanoparticles (SeNPs) on the antioxidant status of rats.

Methods

The males of the outbreed rat strain Wistar albino were selected as a model organism. Animals were fed different forms of selenium. The control group was given a mixture without selenium addition, whereas other groups were fed a mixture containing sodium selenite, Se-49, and Se-100 SeNPs respectively. The duration of the trial was 30 days.

Results

Analysis of blood and liver was performed where the concentration of reduced (GSH) and oxidised (GSSG) glutathione, and total selenium content were measured. In the liver, a significant reduction in GSSG was found for all experiment groups. Blood samples showed a significant reduction in GSH and an increase in GSSG.

Discussion

These results show that SeNPs may be an alternative to dietary selenium for animal organisms.

Introduction

The antioxidant status of animals can be positively affected by the addition of antioxidants, including vitamin E and antioxidant enzyme cofactors such as selenium, which is an important element in selenoproteins, of which at least 16 have an antioxidant role. Interaction between Se and Vitamin E may increase the production of glutathione peroxidase, which is an important aspect of the antioxidant system (Arruda et al., 2015; Horky et al., 2016b; Chen et al., 2016a, 2016b; Skalickova et al., 2017; Tran & Webster, 2011; Wang, Zhang & Yu, 2007; Zhang et al., 2001). Moreover, selenium supports immune response, where in enzyme deiodinase it is necessary for conversion of thyroxine (T4) to more active triiodothyronine (T3) (Bunglavan et al., 2014).

The selenium content in soils in Europe is generally low and as such, it should be added to livestock feed (Horky et al., 2012; Kursa et al., 2010). The two most widely used inorganic selenium forms are selenate and selenite. Both can be converted into less toxic insoluble elemental selenium forms. However, the biological nature of this reaction is not yet known (Chen et al., 2016b). In organic form, selenium is used as a component of amino acids (e.g. selenomethionine) (Horky et al., 2013; Mohapatra et al., 2014). Selenium in a low dose is an essential element important in several physiological processes such as synthesis of selenocysteine, coenzyme Q, glutathione peroxidase, and thioredoxin reductase. At higher doses, selenium may be toxic (Fernández-Llamosas et al., 2016; Horky, 2014).

Accordingly, alternative nanotechnological solutions are required, as opposed to conventional alternatives. In this context, nanoparticles indicate new and promising properties that can potentially suppress toxicity, while maintaining the positive effects of selenium on an organism (Arruda et al., 2015; Fernández-Llamosas et al., 2016; Horky et al., 2012; Mohapatra et al., 2014; Skalickova et al., 2017). The synthesis and application of selenium nanoparticles (SeNPs) has gained increased attention due to the number of benefits it presents, such as low toxicity, biocompatibility, and chemical stability (Zhang et al., 2001). Nowadays, SeNPs are widely used as a nutritional supplement (Wang, Zhang & Yu, 2007). SeNPs have been found to show lower cytotoxicity compared to inorganic selenium compounds, and have excellent anti-cancer and therapeutic properties (Anjum et al., 2016). Zhang, Wang & Xu (2008) showed that SeNPs exhibited novel in vitro and in vivo antioxidant activities using the activation of selenoenzymes. On the other hand, the antiproliferative activity of these nanoparticles according to an unknown mechanism have also been identified (Peng et al., 2007), as well as their antimicrobial effects (Hegerova et al., 2017; Tran & Webster, 2011).

The aim of our study was to compare two different forms of dietary nanoselenium with sodium selenite to show whether SeNPs can increase the antioxidant status of rat metabolism, and serve as an alternative source of nutritional supplement for an animal organism.

Materials and Methods

Animals

The experiments were performed with the approval of the Ethics Commission at the Faculty of AgriSciences, Mendel University in Brno, Czech Republic (project number 02154869). The experiment was carried out in the experimental facility of the Department of Animal Nutrition and Forage Production of Mendel University in Brno, in accordance with Act No. 246/1992 Coll. on the protection of animals against cruelty. Throughout the entire experiment, microclimatic conditions were measured and controlled at 23 ± 1 °C, and at a constant humidity of 60%. The light regime was maintained at 12 h of light and 12 h of darkness, with a maximum illumination of 200 lx.

Laboratory male rats of the outbreed strain Wistar albino were selected as model animals, and included 32 rats aged 28 days, with an average initial weight of 150 ± 5 g. The rats were divided into four groups of eight rats each. The first group served as a control with no addition of selenium to their feed. The second group was supplemented with selenium in the form of Na2SeO3 at a dose of 1.2 mg/kg/diet. Group three and four were fed with selenium in the form of Se-49 and Se-100 nanoparticles at a dose of 1.2 mg/kg/diet respectively. Groups two, three, and four were fed monodietus containing 0.03 mg Se/kg/diet. The experiment duration was 30 days. The animals had access to feed and drinking water ad libitum. At the end of the experiment, the animals were putted to death and blood and liver samples were collected and subjected to chemical analyses.

Chemicals and instruments

Methanol, trifluoroacetic acid (TFA), sodium selenite, poly(vinyl alcohol) (PVA 49 kDa or PVA 100 kDa), reduced glutathione (GSH), and oxidised glutathione (GSSG) were obtained from Sigma-Aldrich (St. Louis, MO, USA) in ACS purity, unless noted otherwise. Deionised water underwent demineralisation by reverse osmosis using the instrument Aqua Osmotic 02 (AquaOsmotic, Tisnov, Czech Republic), and was subsequently purified using Millipore RG (18 MΏ; Millipore Corp., Billerica, MA, USA) to gain MilliQ water. The average particle size distribution was determined by quasi-elastic laser light scattering using a Malvern Zetasizer (NANO-ZS; Malvern Instruments Ltd., Worcestershire, UK). Solutions of nanoparticles were measured according to experimental conditions stated in (Dostalova et al., 2016). The structures of nanoparticles were observed using scanning electron microscopy (FE Tescan Mira II LMU, Brno, Czech Republic) under the conditions employed in (Dostalova et al., 2016; Chudobova et al., 2014). Characterisation of nanoparticles is shown in Fig. 1.

Figure 1 Characterisation of nanoparticles.

(A) Hydrodynamic diameter distribution of nanoselenium particles Se-49 measured by quasi-elastic laser light scattering with a Malvern Zetasizer. (B) shows the SEM image of Se-49 obtained from FE Tescan Mira II LMU. (C) Hydrodynamic diameter distribution of nanoselenium particles Se-100. (D) shows SEM image of Se-100.

Preparation of selenium nanoparticles

Se-49

Poly(vinyl alcohol) 49 kDa (0.19 g) was added to a solution of 1.88 mL Na2SeO3·5H2O (2.63 g/50 mL) in water (80 mL). Cysteine (9 mg/mL) was added with mixing, and left for 2 h. At this stage, the colour turned a light orange and water was added to achieve a final 100 mL volume.

Se-100

Preparation was the same as in the previous instance, with the exception of using PVA 100 kDa instead of PVA 49 kDa. Undissolved PVA was filtered off. After the addition of cysteine, the colour turned to orange and water was added to gain a final 100 mL volume.

Preparation of samples for GSH and GSSG detection

Liver: 2 g of samples from each variant were homogenised in a fritted bowl with the addition of liquid nitrogen and 1.5 mL water. Following homogenisation, each sample was sonicated using an ultrasound needle for 2 min, shaken for 10 min, and centrifuged for 20 min at 25,000g, and at 4 °C. Following on, 100 μL of supernatant was taken from each sample and mixed with 100 μL of 10% TFA, and centrifuged again for 20 min at 25,000g and 4 °C. Following centrifugation, the supernatant was taken and analysed by high-performance liquid chromatography with electrochemical detection (HPLC-ED; Fig. 2).

Figure 2 Sample preparation.

Workflow diagram of the experiment. (A) Tissue extraction and blood collecting, (B) liver and blood, (C) tissue and/or blood microwave assisted mineralisation, (D) determination of Se content by AAS and GSH, GSSG content by HPLC-ED. Photos by Zuzana Lackova.

Blood: sample processing was performed by pipetting 200 μL of sample from each variant, placing it into liquid nitrogen for 2 min and adding 500 μL water. Each sample was sonicated with an ultrasound needle for 2 min, shaken for 1 min, and centrifuged for 20 min at 25,000g and at 4 °C. Then, 200 μL of supernatant was taken from each sample and mixed with 200 μL of 10% TFA. The samples were again centrifuged for 20 min at 25,000g and 4 °C. Following centrifugation, the supernatant was analysed by HPLC-ED (Fig. 2).

Preparation of samples for selenium detection

Samples of liver weighing 0.3 g and samples of blood weighing 0.5 g were disintegrated by dry method in a muffle furnace (LAC, Rajhrad, Czech Republic) and mineralised in 2.5 mL concentrated nitric acid (Horky et al., 2016a). The preparation scheme is shown in Fig. 2.

Determination of reduced and oxidised glutathione and selenium

Reduced and oxidised glutathiones were determined using HPLC-ED. Experimental conditions were adopted from (Zitka et al., 2012). Selenium was determined using a 280Z Agilent Technologies atomic absorption spectrometer (Agilent, Santa Clara, CA, USA) with electrothermal atomisation under the conditions stated in (Horky et al., 2016b).

Statistics

The data were processed statistically using Statistica.Cz, version 10.0 (Czech Republic). The three measurements were taken, P < 0.05 was considered significant using ANOVA, and Scheffe’s method was used determining parameters GSH, GSSG, and Se.

Results

In the experiment, conventional (sodium selenite) and alternative (SeNPs) forms of selenium as sources of this element for animal organisms were investigated. Oxidative glutathione, oxidised glutathione, and selenium in blood and liver were selected as markers of oxidative stress. The level of oxidised and reduced glutathione showed a smaller increase in liver and blood, with the exception of GSH in liver samples. In the liver tissue, a significant decrease of 30% was found in the Na2SeO3 group (P < 0.05), together with both groups containing SeNPs, Se-49 at 34% (P < 0.05), and Se-100 at 29% (P < 0.05) (Fig. 3A). In blood, a statistically significant reduction in GSH was found for all control groups (Na2SeO3 by 72%, Se-49 by 59%, Se-100 by 67%; P < 0.05). Conversely, a 17% increase in GSSG was found in the Na2SeO3 group (P < 0.05). In the case of nanoparticle treated experimental groups, Se-49 increased by 51% (P < 0.05), and Se-100 by 47% (P < 0.05) (Fig. 3B).

Figure 3 Glutathiones.

Influence of different forms of selenium on the level of GSH a GSSH in (A) liver and (B) blood.

Additionally, we also determined selenium content. In liver samples, a significant increase of 85% was observed in selenium concentration in the Na2SeO3 group (P < 0.05), an increase in Se-49 by 30% (P < 0.05), and in Se-100 by 73% (P < 0.05), compared to rats in the control group (Fig. 4A). The level of selenium in blood was also the highest in the Na2SeO3 group, where there was an increase of 240% (P < 0.05) against the control group. Other groups showed significant increase as well: Se-49 by 18% (P < 0.05) and Se-100 by 64% (P < 0.05) (Fig. 4B).

Figure 4 Selenium.

Effect of different forms of selenium on concentration of selenium in (A) liver and (B) blood.

Discussion

In our experiment, the effect of an alternative source of selenium, i.e. SeNPs, was studied in terms of influencing the antioxidant potential of a rat organism. Antioxidant activity is an indicator of the ability of the entire body and selected organs to defend against free radicals. Reducing the antioxidant activity of the organism leads to an intensification of oxidative stress that affects the entire body, increases the risk of injury, reduces performance, and deteriorates certain diseases.

At present, there is relatively little existing work on the use of nanoselenium in diet. In a study on the reduction of radioactive gamma radiation, selenium particles were given at a dose of 20 mg Se/kg of body weight per day (i.e. 3 mg Se/animal/day), and 0.1 mg Se/kg of body weight per day (0.015 mg Se/animal/day). The level of selenium and GSH was not affected (El-Batal et al., 2012). In contrast, the selenium level was increased by 64% in the Se-100 group and GSH level decreased in all our experimental groups in blood samples. However, it should be noted that the animals had not been exposed to gamma rays, which will undoubtedly affect animals’ antioxidant status.

The effect of SeNPs applied to a sugar carrier (glucose) was studied (Horky et al., 2016a). Selenium particles were given at a dose of 0.02 mg Se/animal/day. After 10 days, an increase in GSH and total GPx activity in blood was found, which is inconsistent with our trial, in which GSH elevation did not occur. In another experiment on rats (Hadrup et al., 2016), the effect of selenium and selenium nanocomponents’ addition (0.05 mg/kg bw and 0.5 mg/kg bw) was compared with a control group. The doses were added to feed as solutions using a gastric tube every other day, and urine samples were collected. After 14 days, no toxic effects and no evidence of weight reduction were observed, compared to the control.

In the past, rat experiments were conducted to compare the effect of organic and inorganic selenium. According to (Sochor et al., 2012), the addition of 1.5 mg of Se in organic form (yeast) increased GSH and GPx activity, when compared to sodium selenite. It follows from the results found that the addition of 1.5 mg may increase the antioxidant potential of animals, without the occurrence of signs of toxicity. Another group of authors (Kominkova et al., 2015) indicate the optimal amount of GSH and GSSG as 90% or 10% respectively. In our experiment, higher levels of GSSG (oxidised form) were observed in all selenium addition groups. In blood, the difference was the most significant. It is possible that our selected amount and form of selenium (1.2 mg/kg diet) already had depression in the optimal GSH:GSSG ratio. However, our results correspond to results in a study (Bláhová et al., 2014), where measured concentrations in the liver ranged from 6 to 800 nmol/g for GSH, and from 30 to 800 nmol/g for GSSG. Similar results for the liver were also recorded in a study (Guan et al., 2003). For blood samples, we achieved a higher concentration of GSH and GSSG than in (Guan et al., 2003; Horky et al., 2016a); this difference was most likely caused by another approach to sample preparation and the analysis itself.

Conclusion

The experiment investigated the effect of SeNPs on the antioxidant status of laboratory rats. Alterations in reduced and oxidised glutathiones revealed marked changes in antioxidant status-based selenium treatment; however, we confirmed that the nano-form of selenium has fewer negative effects than the standard form. This leads us to support the idea of using nanoSe as an alternative source of selenium. Moreover, the possibilities of various modifications of the surface of particles is an additional advantage of using these particles, rather than standard inorganic forms, as we show that such nanoSe can be employed without harming animals. It will be appropriate to test these selenium sources at even lower concentrations in order to avoid potential toxicity.

Supplemental Information

Supplemental Information 1 Raw data.

Click here for additional data file.

Additional Information and Declarations

Competing Interests

Author Contributions

Animal Ethics

Data Availability

The authors declare that they have no competing interests.

Lenka Urbankova performed the experiments, prepared figures and/or tables, authored or reviewed drafts of the paper, approved the final draft.

Pavel Horky conceived and designed the experiments, analysed the data, authored or reviewed drafts of the paper, approved the final draft.

Jiri Skladanka conceived and designed the experiments, analysed the data, contributed reagents/materials/analysis tools, authored or reviewed drafts of the paper, approved the final draft.

Magdalena Pribilova performed the experiments, authored or reviewed drafts of the paper, approved the final draft.

Vendula Smolikova performed the experiments, authored or reviewed drafts of the paper, approved the final draft.

Pavel Nevrkla performed the experiments, analysed the data, prepared figures and/or tables, authored or reviewed drafts of the paper, approved the final draft.

Natalia Cernei performed the experiments, prepared figures and/or tables, authored or reviewed drafts of the paper, approved the final draft.

Zuzana Lackova performed the experiments, prepared figures and/or tables, authored or reviewed drafts of the paper, approved the final draft.

Josef Hedbavny performed the experiments, authored or reviewed drafts of the paper, approved the final draft.

Andrea Ridoskova analysed the data, prepared figures and/or tables, authored or reviewed drafts of the paper, approved the final draft.

Vojtech Adam conceived and designed the experiments, contributed reagents/materials/analysis tools, authored or reviewed drafts of the paper, approved the final draft.

Pavel Kopel conceived and designed the experiments, contributed reagents/materials/analysis tools, authored or reviewed drafts of the paper, approved the final draft.

The following information was supplied relating to ethical approvals (i.e. approving body and any reference numbers):

The experiments were performed with the approval of the Ethics Commission at the Faculty of AgriSciences, Mendel University In Brno, Brno, Czech Republic (project Number 02154869).

The following information was supplied regarding data availability:

The raw data are provided in the Supplemental File.

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
