# Peer review of "Antioxidant status of rats’ blood and liver affected by sodium selenite and selenium nanoparticles"

_PeerJ, doi:10.7717/peerj.4862_

## Round 0.1 · original submission · Major Revisions

Please address all critical issues raised by both reviewers.

Reviewer 1 ·

Basic reporting

The authors of this manuscript describe the antioxidant status in blood and liver of rats treated with sodium selenite or selenium nanoparticles. The aim of the study is to compare two different forms of selenium such as nanoselenium particles with sodium selenite to show whether selenium nanoparticles can increase the antioxidant status of rat metabolism and serve as alternative nutritional supplements for an animal organism.

Experimental design

The results are summarized in the well-arranged schema and graphs that offer detailed information. All experiments were very well designed and executed. The manuscript is clearly written and easy to follow. Therefore I have only a few points.

Validity of the findings

The conclusions of the study are clearly stated but the lack of knowledge on the toxic effect of nanoparticles in animals is an issue.
The hemolysis assay could be added to verify the toxic effect of nano-particles on mammalian cells.

Additional comments

I would suggest a minor correction of the English language style.

The background of the abstract does not have sufficient information about the need for new forms of selenium with lower toxicity as nutritional supplements.

The conclusion needs to be rewritten and it should clearly stay the beneficial effects of nano-particles over sodium selenite.

Thyroxine is mistyped at line 37.

The gender and age of rats should be mentioned in the Materials and Methods section.

The water content in the treated and untreated animals could have a big variation. I would suggest verifying the weight of livers with the protein concentrations in the extracts. The same amount of protein in the extracts should be analyzed and compared.

Reviewer 2 ·

Basic reporting

The manuscript is written well.

Experimental design

More experiments need to be done. Please see attached comments.

Validity of the findings

Work is not conceptually novel.

Additional comments

In this paper Urbankova and colleagues synthesized two selenium nanoparticles (Se-NPs), Se-49 and Se-100 and studied their antioxidant status on rat by measuring reduced (GSH) and oxidized (GSSG) glutathione level, and the total selenium content in blood and liver samples. Authors observed differential effects on GSH, GSSG and selenium level in liver and blood—and based on their observations authors concluded that Nano-form of selenium has better antioxidant property compared to sodium selenite, a conventional dietary supplement.

The idea of using Se-NPs as alternative dietary supplements is evolutionary. There is numerous numbers of studies with different Se-NPs to characterize their effects on living organisms and to understand whether Se-NPs could be used as an alternative of conventional dietary Se-supplements. Therefore, the current study is not completely new except using of two new Se-NPs and studying their effects on rat. This reviewer does not think that the work is complete enough to publish in PeerJ in its current shape, however can be consider after answering the following concerns.

1. Several other important experiments must be done in order to characterize a new Se-NP comprehensively, such as 1) activity measurement of Se-dependent enzymes, 2) measurement of markers for liver injury, 3) activity assay of liver GST.

2. Please check the existence of this reference: Sochor J, Pohanka M, Ruttkay- Nedecky B, Zitka O, Hynek D, Mares P, Zeman L, Adam V, and Kizek R. 2012. Effect of selenium in organic and inorganic form on liver, kidney, brain and muscle of Wistar rats. Cent Eur J Chem 10:1442-1451. 10.2478/s11532-012-0064-8.

3. In line 177-179: authors conclude the GSSG level goes up significantly in blood upon selenium treatment. It would be more convincing if authors calculate and provide “p-values”. In addition, it is also suggested to calculate and plot the ratio of reduced GSH to GSSG.

4. This reviewer also recommends replicating the studies with different doses of Se-NPs in diet and after different time intervals from the day of treatment begins.

Please see these comments in the attached PDF

Annotated reviews are not available for download in order to protect the identity of reviewers who chose to remain anonymous.

---

## Round 0.2 · accepted · Accept

Thank you very much for addressing critical issues raised by the reviewers and for the careful revision. The revised paper was strengthened and therefore is accepted for publication.

# Reviewer 2 ·

Basic reporting

No comment

Experimental design

No comment

Validity of the findings

No comment

Additional comments

The authors have addressed my concerns. The revised paper strengthens the few weak points in the earlier version and is ready for publication.